# Novel LiAlO_2_ Material for Scalable and Facile Lithium Recovery Using Electrochemical Ion Pumping

**DOI:** 10.3390/nano13050895

**Published:** 2023-02-27

**Authors:** Tasneem Elmakki, Sifani Zavahir, Umme Hafsa, Leena Al-Sulaiti, Zubair Ahmad, Yuan Chen, Hyunwoong Park, Ho Kyong Shon, Yeek-Chia Ho, Dong Suk Han

**Affiliations:** 1Center for Advanced Materials, Qatar University, Doha P.O. Box 2713, Qatar; 2Department of Mathematics, Statistics and Physics, College of Arts and Sciences, Qatar University, Doha P.O. Box 2713, Qatar; 3Qatar University Young Scientists Center (QUYSC), Qatar University, Doha P.O. Box 2713, Qatar; 4School of Chemical and Biomolecular Engineering, The University of Sydney, Sydney, NSW 2006, Australia; 5School of Energy Engineering, Kyungpook National University, Daegu 41566, Republic of Korea; 6School of Civil and Environmental Engineering, Faculty of Engineering and IT, University of Technology Sydney, P.O. Box 123, Broadway, NSW 2007, Australia; 7Centre for Urban Resource Sustainability, Institute of Self-Sustainable Building, Universiti Teknologi PETRONAS, Seri Iskandar 32610, Malaysia; 8Department of Chemical Engineering, College of Engineering, Qatar University, Doha P.O. Box 2713, Qatar

**Keywords:** α-LiAlO_2_, activated carbon, lithium recovery, electrochemical ion pumping

## Abstract

In this study, α-LiAlO_2_ was investigated for the first time as a Li-capturing positive electrode material to recover Li from aqueous Li resources. The material was synthesized using hydrothermal synthesis and air annealing, which is a low-cost and low-energy fabrication process. The physical characterization showed that the material formed an α-LiAlO_2_ phase, and electrochemical activation revealed the presence of AlO_2_* as a Li deficient form that can intercalate Li^+^. The AlO_2_*/activated carbon electrode pair showed selective capture of Li^+^ ions when the concentrations were between 100 mM and 25 mM. In mono salt solution comprising 25 mM LiCl, the adsorption capacity was 8.25 mg g^−1^, and the energy consumption was 27.98 Wh mol Li^−1^. The system can also handle complex solutions such as first-pass seawater reverse osmosis brine, which has a slightly higher concentration of Li than seawater at 0.34 ppm.

## 1. Introduction

Lithium is one of the most sought after elements in industry due to its use in high-power, single-use or rechargeable Li-ion batteries for portable electronics, precision electronics [1], and electrical vehicles (EV) [2,3]. The market demand for Li for the energy industry alone has risen from 35% to 74% [4], causing a demand-driven price increase of battery-grade Li_2_CO_3_ from 8000 USD per metric ton in 2019 to a record-high 17,000 USD in 2020. In addition, Li is also widely used in the manufacture of ceramic and glass [5], aluminum smelting, pharmaceuticals [6,7], Weldalite Al–Li binary alloy for the aerospace industry [8,9], and as a pore-forming agent in the manufacture of porous polymers [10]. Hence, the global market demand for Li is growing faster than ever. According to the USGS, a total of 89 Mt of Li is available in various forms, such as liquid brines and solid or molten minerals, as of 2022 [11].

Li-containing minerals, which are part of the primary Li resource along with brine and seawater, are extracted using mature industrial techniques. Although Li can be found in 145 different minerals, only the first five minerals, such as spodumene, lepidolite, petalite, amblygonite, and eucryptite, have been targeted for extraction due to their easy access, high proportion of Li, and relative hardness. Liquid Li resources make up two-thirds of all forms of Li resources [12,13]. The extraction of Li from brines involves many steps, with the first step being the evaporation of brine in large evaporation ponds using wind and solar power. There are eight active facilities globally, two each in Argentina and Chile, three in China, and one in the US. These facilities are mainly centered around salt lake brines, which have a diverse yet high concentration of Li ranging from 100–1000 ppm, compared to the very low concentration of Li in seawater (0.17 ppm). The drying process usually continues until the concentration of Li reaches around 5000 ppm, after which the brine is moved to a recovery pond. The exact steps depend on the type of brine. Although solar drying seems straightforward and cost-effective, it is challenging to implement due to geological and climatic constraints. In an arid and windy environment, the brine is concentrated over 24 months to achieve the 5000 ppm Li level. It takes processing 500,000 L of brine to extract 1 ton of Li_2_CO_3_ [14]. Efforts are being made to find alternative methods of enriching Li to address the time-consuming nature of aquifer depletion and standardize the process.

There have been both passive and electrochemical methods developed to improve the extraction of Li from brine [14,15,16]. Ionic sieve-type adsorbents have proven to be highly efficient, offering clean processing and high-quality Li recovery [17]. Lithium-specific ion sieves, commonly synthesized using aluminum oxide, lithium manganese oxide, and lithium titanium oxide, function through memory effect and ion screening [18,19] and are designed to selectively extract Li from the final structure. However, the ion sieve method also has its challenges. The process is slow as it relies on thermodynamic equilibrium and takes a significant amount of time to reach it. Additionally, the rate of Li diffusion in aqueous solutions is low, and the ion exchange rate in the adsorbents is moderate, further hindering the process [20,21]. On the other hand, Li concentration using electrochemistry is primarily driven by a redox process or electrode polarization under a current flow, accompanied by changes in the electric field at the interface [22,23,24]. Unlike the adsorption process, electrochemical methods do not require the heavy use of chemicals to elute Li from the electrode [16]. Instead, Li can be concentrated in a recovery solution by reversing the current flow during capture [24,25,26]. Electrically switched ion exchange (ESIX) for Li concentration is becoming increasingly prominent due to the high-performance matrices produced by the technique. It features higher Li removal capacities, efficiencies, and minimal water usage in both the capture and release process [27].

The ESIX process is effectively used to separate target ions from aqueous solutions by controlling the voltage profile or current density. This approach has been applied to recover or remove Ag^+^ [28], K^+^ [29], and Ce^+^ [30] ions from wastewater, enhancing the potential for recycling and reuse of water. Manganese-based spinel electrodes were initially explored for selective capture of Li from Li salt solutions with a Pt counter electrode by sweeping the potential region from 0.246 V to 1.244 V [31]. The working principle of ESIX is similar to that of battery cycling. The Li capture-release cycles in ESIX use aqueous electrolytes as the source and recovery solutions, which provides a high level of safety due to the limited operational potential window [32,33]. Building on the work of Kanoh et al., many studies have been conducted on Li-selective positive electrodes and negative electrodes. The ƛ-MnO_2_, which originated from delithiated LiMn_2_O_4_ with a cubic spinel structure and the fd3-m space group, has tetrahedral 8a sites for Li and 16d octahedral sites for Mn. The ƛ-MnO_2_-Ag combination as a Li-capturing positive electrode and Cl-capturing negative electrode pair showed high Li capture in the presence of Mg and other competing cations in an energy-efficient process, consuming 1 Wh mol^−1^ [23]. The ƛ-MnO_2_-Ag and ƛ-MnO_2_-NiHCF systems, when studied with Ag as the Cl^−^ capturing electrode, showed high stability of the ƛ-MnO_2_ positive electrode, which could be cycled over 100 cycles without significant loss in specific capacity, 100 mAh mol^−1^ [34]. When paired with a capacitive activated carbon negative electrode, the ƛ-MnO_2_ electrode achieved a Li^+^ ion concentration increase of 19 mM per cycle with a low energy consumption of 4.2 Wh mol^−1^, a comparable level to that of ƛ-MnO_2_-Ag [26]. The ƛ-MnO_2_ or (activated/delithiated) LiMn_2_O_4_ systems recovered Li in the range of 6.93 to 31.8 mg per gram of active manganese material in the positive electrode [31,35,36,37]. The LiFePO_4_–Ag electrode couple in contemporary studies showed a greater extent of Li concentration, converting a sodium-rich brine (Li:Na = 1:100) into a Li-rich brine (Li:Na = 5:1) with a calculated energy consumption of 144 Wh kg^−1^ of Li recovered [24].

The LiMn_2_O_4_ and LiFePO_4_ materials show remarkable performance characteristics and good environmental compatibility [38]. However, due to the scarcity of raw materials, the high cost of the active material, and the long and complicated preparation processes, it is important to search for alternative positive electrode materials that are abundant, inexpensive, and easy to synthesize. Aluminum oxide-based ionic sieves have been used in the Li selective membrane process, but LiAlO_2_ continues to be studied as a topcoat or support layer to increase the ionic conductivity, structural stability, and cycling performance of battery materials [39,40,41,42]. It is surprising that LiAlO_2_ is not considered as a Li-capturing positive electrode material in the electrochemical ion pumping process despite computational studies showing its potential based on thermodynamics, ionization of Li, and the charge transfer between metal and anions along with the density of states [43]. This study aims to investigate the possibility of LiAlO_2_ as the primary electrode material for capturing and concentrating Li using ion pumping technology for the first time as the demand for Li continues to grow. Our experiment explores the practical limits of LiAlO_2_ as a sustainable and powerful Li capture electrode candidate, which can potentially be scaled up to an industrial level.

## 2. Materials and Methods

### 2.1. Materials

The following chemicals were purchased from Sigma Aldrich, St. Louis, MO, USA: Li_2_CO_3_, Al_2_O_3_, LiCl, KCl, N,N-poly(vinylidene fluoride) PVDF, and N-methyl pyrrolidone (NMP). The acetylene carbon was purchased from MTI company, Richmond, CA, USA. All chemicals were used as received without further purification. Deionized water (DI) was used from a Milli Q ultra-purification system (type 2) with a resistivity of 18.2 MΩ cm^−1^ m, where necessary.

### 2.2. LiAlO_2_ Preparation

α-LiAlO_2_ was synthesized using a hydrothermal process. A solution (100 mL) of Li_2_CO_3_ (0.05 mol) and Al_2_O_3_ (0.05 mol) was made and transferred to a 500 mL Teflon liner. The solution was then autoclaved at 200 °C for 16 h in air. The white solid produced was separated from the mother liquor, washed with DI water three times using centrifugation (at 4000 rpm for 10 min per cycle), dried overnight at 110 °C in a vacuum oven, and then finely ground using a mortar and pestle. The resulting powder was then placed in a quartz crucible and annealed at 720 °C for 8 h with a heating rate of 10 °C per min.

### 2.3. Electrode Preparation

Fluorine-doped tin oxide (FTO) glass was utilized as the substrate for the electrode material. Pieces of FTO glass measuring 3 cm × 2 cm were thoroughly cleaned using ultrasonic waves with ethanol, acetone, and DI water, each for 5 min. The electrode coating was then drop-cast onto the conducting surface of the cleaned FTO glass substrate. The conducting side of the FTO glass was determined using a multimeter. The casting solution was prepared by combining active component (LiAlO_2_), PVDF, and acetylene carbon in a weight ratio of 80:10:10. 600 µL of NMP was added to create a slurry of the desired consistency, which was then drop-cast onto a 2 cm × 2 cm area. The coat was dried in a vacuum oven preheated to 70 °C for 3 h. The positive electrode, ready for use in the electrochemical ion pumping cycle, had a real mass of 70–90 mg cm^−2^ with respect to the active material.

### 2.4. Material Characterization

The surface morphology of the electrode material was characterized using a scanning electron microscope (SEM) equipped with an energy dispersive X-ray (EDX) spectroscopy (SEM/EDX, NOVANANOSEM 450, FEI company, Hillsboro, OR, USA). The crystallographic pattern of the synthesized electrode material was analyzed using an X-ray diffractometer (XRD, PANalytical Empyrean, Malvern Panalytical, Malvern, UK) with a Cu Kα (ƛ = 1.5406 Å) X-ray source, operating at 40 kV and 30 mA current, and equipped with a solid-state detector. The lithium concentration in liquid test samples from batch experiments in discharge and charge cycle was measured by an inductively coupled plasma (ICP)-optical emission spectrometer (OES) instrument (ICP5000 Dual View, PG Instruments, Lutterworth, UK) after calibration with standard solutions of LiCl.

### 2.5. Electrochemical Characterization

The electrochemical behavior of the LiAlO_2_-based electrode was characterized using cyclic voltammetry (CV) tests, which were performed in three-electrode assembly on Gamry 5000 interface (Gamry Instruments, Warminster, PA, USA) potentiostat using a 1 M LiCl solution and 1 M Na_2_SO_4_. A Pt wire and Ag/AgCl were used as the counter and reference electrodes, respectively.

### 2.6. ESIX Li Capture–Release Cycling Tests

The galvanostatic charging/discharging (GCD) tests were conducted in a two-electrode system using a solution containing 0.1 M LiCl, and the LiAlO_2_ and activated carbon (AC) film-coated electrodes were used as the positive and negative electrodes, respectively. A cycling test of the ESIX system was performed at a current density of ±0.5 mA cm^−2^. The Li-ion concentration effect was studied using 5, 25, 50, and 100 mM LiCl source solutions. The recovery solution was 50 mM KCl. The effect of multi ions was studied with dual salt solutions containing LiCl–NaCl and LiCl–MgCl_2_. Before each discharge experiment, the LiAlO_2_ electrode was activated by applying a positive current density of 0.2 mA cm^−2^ for 2 h in 1 M LiCl solution.

### 2.7. Stability Tests

The long-term durability of the electrode was assessed through continuous operation in ESIX cycling for 10 cycles. After each discharge–charge cycle, the electrode was immersed in a 0.1 M HCl solution for 4 h and then washed with DI water and air-dried under normal atmospheric conditions.

## 3. Results and Discussion

### 3.1. α-LiAlO_2_ Material Characterization

The scanning electron microscopic (SEM) images of LiAlO_2_ in Figure 1a display a structure with a long-range ordered arrangement of approximately 400 nm layer thickness. The image shows that the regular arrays of LiAlO_2_ bundles have random orientation in various colonies. Figure 1b, which is a TEM image focusing on a single-layer unit, depicts the aggregation of nanosized particles into larger structures. The selected area diffraction pattern (SAED) provided in the inset of Figure 1b reveals scattered diffraction points at different radii, indicating weak crystallinity. The powder XRD analysis confirmed the formation of the α-LiAlO_2_ phase of the material. As depicted in Figure 1c, the prominent 2θ diffraction angles located at 18.71°, 37.59°, 39.22°, 45.48°, 49.35°, 58.37°, 59.30°, 65.02°, and 66.76° are similar to (003), (101), (012), (104), (015), (009), (107), (018), and (110) planes of α-LiAlO_2_, respectively. α-LiAlO_2_ has a hexagonal crystal system with an R3m space group. Additionally, peaks related to the γ-LiAlO_2_ phase were also observed in Figure 1c, indicating the existence of mixed phases of α-LiAlO_2_ and γ-LiAlO_2_. LiAlO_2_ electrodes were prepared with the annealed material on FTO glass using acetylene carbon and PVDF powder and were tested in 1 M Na_2_SO_4_ and 1M LiCl electrolyte solution at a sweep rate of 5 mV s^−1^. The experiment showed a higher capacitance with the 1 M LiCl electrolyte, but not with 1 M Na_2_SO_4_, demonstrating greater Li^+^ ion conductivity and affinity by the LiAlO_2_ cathode material compared to the Na^+^ ion, as seen in Figure 1d [44].

### 3.2. Li Recovery Cycle of LiAlO_2_ and AC Electrode Pair

This study focuses on the application of LiAlO_2_ as a Li-capturing cathode in a four-step Li recovery cycle using electrochemical ion pumping. The Li capturing electrode used is a delithiated (activated) LiAlO_2_ electrode (AlO_2_*), while the counter electrode is a commercial AC electrode with an anion exchange membrane. The goal is to evaluate the suitability of LiAlO_2_ as the cathode material. The use of AC offers benefits such as electrical conductivity, low cost, high specific surface area, and environmental friendliness. As can be seen in Figure 2, during step one, the electrode pair is connected to the Li-rich brine solution, and the Li-capturing process takes place under a constant negative current (or constant potential mode) for a specific duration, which is comparable to the discharging cycle of a battery. At this stage, applying a negative current causes Li^+^ ions from the brine solution to be incorporated into the Li voids in the delithiated LiAlO_2_ (AlO_2_*) structure (reaction (1)). Anions are captured by the AC negative electrode, unlike the Li insertion mechanism in AlO_2_* where they are captured on the surface of the AC by physical adsorption.

In step 2, the Li-rich brine solution is replaced with a recovery solution (50 mM KCl) to facilitate the Li^+^ ion concentration process. The concentration is maintained at 50 mM to ensure sufficient ionic conductivity in the electrolyte and minimize ohmic losses.

In the third step, a positive current is applied across the electrodes in the recovery solution. This is the lithium capture step, during which Li is released into the solution while anions adsorbed on the AC are returned to the solution. As Li^+^ ions are released from the LiAlO_2_ electrode, a Li-deficient LiAlO_2_ electrode (technically AlO_2_*) is simultaneously generated, making it ready for another Li capture cycle (Step 4). The cell configuration resembles Step 1, with the difference being that the recovery solution has been replaced with a fresh batch of Li-rich source solution. Energy is consumed during the charging step to release Li ions from the LiAlO_2_ material into the recovery solution.
(1)$${\mathrm{AlO}}^{-}{}_{2}+{\mathrm{Li}}^{+}\to {\mathrm{LiAlO}}_{2}$$

The structure of α-LiAlO_2_ is similar to that of α-NaFeO_2_ or commercially established cathode material, LiCoO_2_ [43] (Figure 3). In this arrangement, oxygen atoms are positioned in a hexagonally stacked close-packing system where cations reside at octahedral sites within the plane of the oxygen layers [41]. Studies of LiCoO_2_ have shown that metal ions Li and (Co or Al) alternate in planes. Thus, removing Li creates greater repulsion on the oxygen planes on either side, leading to contractions in the O–Co–O bonds in LiCoO_2_ [45,46]. In the case of LiCoO_2_, the complete removal of Li occurs without significant distortion of the structure’s symmetry. It is believed that α-LiAlO_2_ exhibits similar behavior to LiCoO_2_ and can therefore undergo reversible Li capture–release cycles without any changes to the material’s overall symmetry.

### 3.3. Selectivity Testing towards Competing Ions

In keeping with our primary objective, we evaluated the potential of the LiAlO_2_ electrode to serve as the Li-capturing electrode, which is typically only used as a filler material in the battery industry, in order to improve ionic conductivity. The exclusive selectivity of the Li-capturing electrode towards Li is of utmost importance in the electrochemical ion pumping process. To test this, three different salt solutions of 100 mM concentration, namely LiCl, NaCl, and MgCl_2,_ were subjected to the described ion capture–release cycle using the electrochemically delithiated LiAlO_2_ electrode as the positive electrode and AC as the negative electrode. NaCl, representing the Na^+^ ion, was chosen because NaCl is the most abundant salt ion that any Li source solution may contain if sourced from a primary liquid Li resource. Mg^2+^ has similar ionic radii to Li^+^, so it was crucial to assess the competitiveness with MgCl_2_. Both the ion capturing and release procedures were performed for 30 min. Figure 4 shows the concentration of the respective metal ions in the source and recovery solutions as the cycle progresses. The batch mode operation results, obtained by testing liquid aliquots collected at 5 min intervals using ICP, demonstrate a significant reduction in the Li concentration in the source solution starting from the initial five minutes, compared to far less concentration of Na^+^ and Mg^2+^ ions in their respective solutions. This can be attributed to the selective insertion reaction between Li^+^ ions in the solution and AlO_2_* units in the electrode, which is not supported in the case of Na^+^ and Mg^2+^. At the end of the discharge step, 22.4 mM of Li ions were inserted into the positive LiAlO_2_ electrode, resulting in an increase in Li-ion concentration in the recovery solution by 20.8 mM. Li^+^ release was proportional to time during the first 20 min of the Li release cycle (step 3), whereas for Na^+^ and Mg^2+^ ions, the concentration decrease in the discharge step and increase in the charge step did not exceed 3 mM after 30 min cycles. The Li concentration efficiency observed in the AlO_2_*/AC system is comparable to that of the ƛ-MnO_2_/AC system studied by Yoon and coworkers [26] in a continuous operation mode in a flow reactor. Overall, 92.8% of the Li captured in the discharge step was released to the recovery solution in the charging step. K^+^, as a competing ion to occupy the voids of the activated AlO_2_, was not examined as the recovery solution used as KCl.

### 3.4. Li Recovery Cycles with Different Initial Li Concentrations

Resources containing Li at a concentration of 100 mM or higher (~690 ppm_Li_) are limited to salt lake brines. Seawater contains the largest amount of Li, at 2 × 10^12^ tons, but at a very low concentration of 0.17 ppm [47]. Secondary Li resources, such as waste from Li batteries, industrial wastewater streams, and seawater reverse osmosis brine, contain Li concentrations starting from 0.34 ppm. Therefore, understanding the Li capture capacity of LiAlO_2_ towards lower Li concentrations is important. LiCl solutions of four different concentrations (5, 25, 50, and 100 mM) discharged at the current density of −0.5 mA cm^−2^ for 90 min and charged at 0.5 mA cm^−2^ for 90 min showed varying degrees of Li concentration. The discharging cycles shown in Figure 5a have a similar shape of the curve with the plateau positioned at different voltages, revealing the influence of Li-ion concentration on the plateau. The potential value at the plateau shifted towards the negative region in accordance with the initial Li-ion concentration, with the order being 100 mM > 50 mM > 25 mM > 5 mM. The thermodynamic and kinetic aspects suggest that more energy is required when the Li^+^ concentration is low. Pasta et al. concluded that at lower Li concentrations (in a system with the co-ion Na^+^), the experimental results are subject to statistics, as the applied current density is much greater than the limiting diffusion current [24]. This reduction in the limiting diffusion current inherently increases the concentration overvoltage, which moves the voltage profile toward negative potential. Although the observed Li capture and release through quantitative detection by ICP, evaluating the mechanism is challenging [48]. A thorough evaluation of the oxide’s structural effects through density functional theory (DFT) indicates that bond relaxation plays a major role in nonrigid band intercalation, with significant charge transfer to the anion upon intercalation of Li into the structure [43].

The corresponding charging cycle (Step 3, Figure 2) at 0.5 mA cm^−2^ for 90 min performed on a 50 mM KCl recovery solution, as shown in Figure 5b, reveals that the Li concentration’s influence on the release cycle is independent of the trend observed in the Li capture cycle in Figure 5a. The voltage plateau was most positive at 1.08 V for 50 mM LiCl. Interestingly, 5 mM LiCl and 100 mM LiCl showed a similar voltage plateau at 0.50 V, and it was further reduced to 0.16 V for Li^+^ released from the 25 mM LiCl initial salt solution. As previously mentioned, the rapid Li concentration change during the charge cycle influences the Li removal voltage from the LiAlO_2_ cage, as supported by the Nernst equation; the potential is proportional to the concentration of electroactive species. To fully understand the Li-ion separation and concentration behavior of AlO_2_* by ion pumping, the subsequent recovery rate (RR) and energy cycle were thoroughly evaluated. Li^+^ ion concentration in the recovery solution after 180 min of complete cycle operation (capture-release) was analyzed using ICP.

Figure 5c demonstrates the influence of the initial Li concentration on the extracted amount of Li and recovery rate. The amount of extracted Li (*w_Li_*, mg) is calculated based on the concentration of Li in the recovery solution (*C_Li_*, mol L^−1^) determined by ICP and the volume of recovery solution (*V_r_*, in mL), as shown in Equation (2). The recovery rate of Li (calculated according to Equation (3)) in relation to the active material in the positive electrode is determined as the mass ratio between the extracted amount of Li (*w_Li_*, mg) and the mass of the active electrode material (*w_e_* in grams; AlO_2_*) in the electrode.
(2)$$w_{Li}=6.9C_{Li}V_{r}$$
(3)RR=wLiwe

The extracted amount of Li is significantly low when the initial Li concentration is 5 mM. However, more Li is extracted when the initial Li concentrations are elevated at 25, 50, and 100 mM. The Li capture rate is similar for 25 mM and 50 mM and seemingly higher for 100 mM solution. Specifically, after 90 min of discharging followed by a charging cycle, 0.99, 6.07, 11.04, and 22.63 mg of Li were extracted. The assessment of the Li recovery rate per gram of the LiAlO_2_ active material in the electrode revealed a recovery rate of 5.05 mg g^−1^ for a 5 mM initial LiCl solution, which is quite low compared to the 8.25 mg g^−1^ recovery rate observed for 25 mM, and a similar rate of 8.5 mg g^−1^ for 50 mM and 100 mM solutions.

In the complete ion pumping cycle (also shown in Figure 2), steps two and four correspond to the change of electrolyte solution from a Li-rich source solution (brine) to a recovery solution and vice versa, respectively, and these steps are purely mechanical. In this study, only the integral area of the potential (Δ*E*)-charge (*q*) curve of the discharge and charge steps were considered for energy calculations. Energy consumption (*W*) related to each cell cycle was calculated using the following equation (Equation (4)).
(4)$$W=\int^{\;}\Delta E.dq$$

Figure 5d shows the voltage profile of the systems in relation to the amount of Li recovered, which is shown in Figure 5c. The voltage change observed in this system is similar to that observed in a previous study by Yoon et al. on the ƛ-MnO_2_/AC system during Li recovery cycles [26]. The voltage drop from 0.8 V to 0.3 V during discharge was linearly followed by an increase to 0.8 V during charging [26], which was attributed to the electric double-layer capacitor property of the AC electrode [49]. However, the energy consumption observed in the ƛ-MnO_2_/AC system was 4.2 Wh mol^−1^ Li, much lower than the energy consumption observed in this study. The normalized energy consumption for one mole of Li recovered was higher in this study, at 27.98 Wh mol^−1^ Li for a 25 mM initial Li concentration solution, 32.15 Wh mol^−1^ Li for 5 mM, and 22.56 Wh mol^−1^ Li for 100 mM and 50 mM. The energy consumption is higher than that reported for most LMO-positive electrodes and AC [26], Ag [23], or NiHCF [32] negative electrode systems. LMO with a boron-doped diamond counter electrode consumed much higher energy at 60.45 Wh mol^−1^ [50]. However, the activation of LiAlO_2_ as Li capturing electrode offers great prospects for improving the LiAlO_2_ system for lower energy consumption and higher Li recovery rates in the future. The higher energy consumption in this study may be due to energy loss associated with concentration overvoltage and ohmic drop.

### 3.5. Li Recovery Cycles with Coexisting Multi-Ions

Individual salt solutions of Na^+^ and Mg^2+^ ions showed no significant ion capture and were released with the activated LiAlO_2_ positive electrode and AC negative electrode (as seen in Figure 4). However, it is important to determine if these ions in the solution can compete with Li^+^ ions and occupy the Li sites if they coexist with the solution. The evaluation of different Li source solution concentrations showed better Li recovery rates at 25 mM, which were comparable to 50 mM and 100 mM, but lower compared to 5 mM. Figure 6a displays the discharge profiles of 25 mM LiCl and the equimolar mixture of LiCl with NaCl and MgCl_2_. The overall shape and plateau of the discharge curve are very similar among the three solution systems. Similarly, the following charging cycles had a similar trend for the mono-salt solution (LiCl) and the multi-salt solutions (NaCl or MgCl_2_), as shown in Figure 6b. This consistency is maintained because there is no other competitive mechanism for inclusion in the background besides Li^+^. The voltage decline from 1.5 V to −0.3 V and subsequent increment to 1.5 V can be seen in the potential profiles of both discharge and charge cycles.

At the end of the complete Li capture–release cycle operation, the samples of the recovery solution were analyzed using ICP, and the concentration of each individual ion and the corresponding selectivity is shown in Figure 6c. After 180 min of cycle operation, 9.1 mM of Li^+^ was recovered along with 0.35 mM of Na^+^ in the LiCl–NaCl mixed salt solution. In the LiCl–MgCl_2_ mixed salt solution, a relatively higher concentration (11.7 mM) of Li^+^ was recovered. Comparing Na^+^ and Mg^2+^ as competing ions, the Li^+^ recovery was 22.2% higher in a mixed solution with Mg^2+^ than in one with Na^+^. Additionally, the capture of Mg^2+^ was only 0.17 mM, which is nearly 50% lower than the capture of Na^+^. The electrode selectivity (*K_Li/M_*) for Li^+^ over co-ions is calculated as the ratio between the Li^+^ (*C_Li_*) concentration after the Li capture–release cycle and the concentration of a competing positive ion (co-ion) in the source solution (*C_M_*) (as per Equation (5)). A higher selectivity coefficient indicates a greater affinity of the active electrode material towards Li^+^ ions than other metal ions. The selectivity factor for Li^+^ with respect to Na is calculated to be 27.8, while it is much higher with respect to Mg^2+^, at 68.7, meaning that Mg^2+^ interferes less with Li^+^ than Na^+^.
(5)KLi/M=CLiCM

### 3.6. Stability Testing

The stability of the electrode was evaluated by performing consecutive 30 discharge–charge cycles. The specific charge of the activated LiAlO_2_ electrode using 25 mM LiCl as the electrolyte showed a value of 27.89 mAh g^−1^. In general, the capacity declined with each cycle. A 2.8% reduction in capacity was observed after the first three cycles, followed by a significant drop from 26 mAh g^−1^ to 18 mAh g^−1^, equivalent to a 35.5% loss of the initial specific capacity from cycle 3 to cycle 8. After cycle 8, the specific charge appeared to stabilize at 17 mAh g^−1^, as shown in Figure 6d. The stability of the electrode during cycling is critical in determining its industrial feasibility. In future work, we plan to study the impact of negative electrodes, particularly in terms of their influence on cycling behavior.

### 3.7. Li Capture from Complex Solution

The LiAlO_2_/AC system was tested for its stability in handling small amounts of Li ions using a simulated solution of 1st pass SWRO brine with a Li concentration of 0.34 ppm (Table 1). Although the concentrations of other cations and anions were measured, they were not taken into account in further calculations. After a complete discharge–charge cycle, the recovery solution was found to have a Li concentration of 78 ppb. Despite the need for further improvement in the LiAlO_2_/AC system, these results suggest the potential of using LiAlO_2_ material as a Li-capturing electrode on a larger scale.

## 4. Conclusions

In this study, we presented the first hybrid supercapacitor system using a LiAlO_2_-based positive electrode and an activated carbon negative electrode. Testing was conducted with salt solutions containing various concentrations of LiCl and multi-ions, as well as simulated SWRO brine, to evaluate the LiAlO_2_ electrode’s capability of capturing and enriching Li in the recovery solution to different levels. In a Li capture–release cycle, 92.8% of Li captured from a 100 mM initial source solution was quantitatively recovered. An equimolar solution of LiCl, NaCl, and MgCl_2_ showed a selectivity ratio (Li/M) of 28 and 69 for Na^+^ and Mg^2+^, respectively, higher than other Li ESIX systems using LiAlO_2_ electrodes. Mg^2+^ showed less competition with Li^+^ than Na^+^. The electrode had an adsorption capacity of 8.25 mg g^−1^ and consumed 27.98 Wh mol^−1^ energy in a 25 mM LiCl source solution. These positive results encourage the further improvement of the electrode pair to reach an environmentally benign and sustainable level for industrial application.

## Figures and Tables

**Figure 1 nanomaterials-13-00895-f001:**
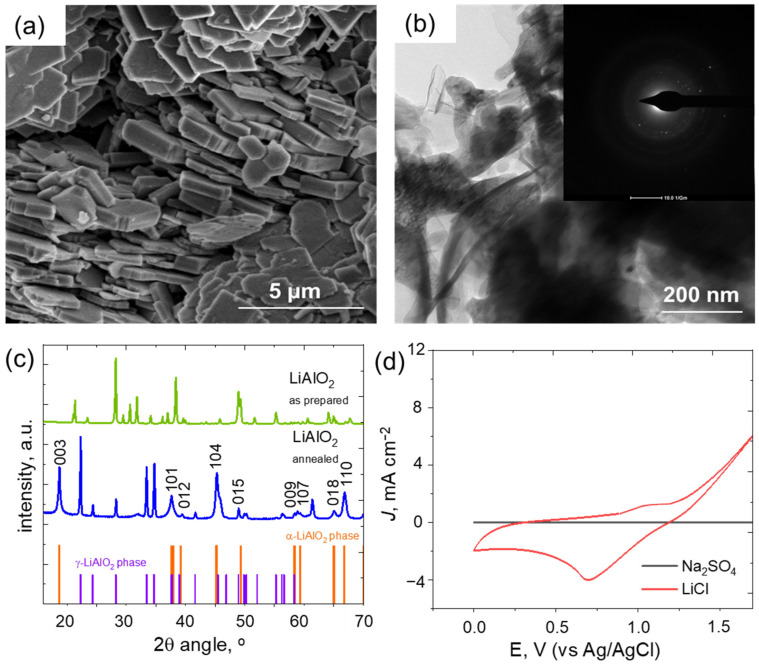
(**a**) SEM image of LiAlO_2_ material with a scale bar of 5 µm. (**b**) TEM image of LiAlO_2_ at a scale bar of 200 nm, with an inset displaying the SAED pattern. (**c**) X-ray diffraction patterns of the as-prepared LiAlO_2_ material and LiAlO_2_ material after annealing. Peaks indexed correspond to the α-LiAlO_2_ phase, with line pattern representing the standard diffraction pattern of α-LiAlO_2_ (long lines, ICDD 98-002-8288) and γ-LiAlO_2_ (short lines, ICDD 98-002-3815). (**d**) *J*–E polarization curve of the LiAlO_2_ measured in 1 M Na_2_SO_4_ and 1 M LiCl electrolyte at a scan rate of 5 mV s^−1^.

**Figure 2 nanomaterials-13-00895-f002:**
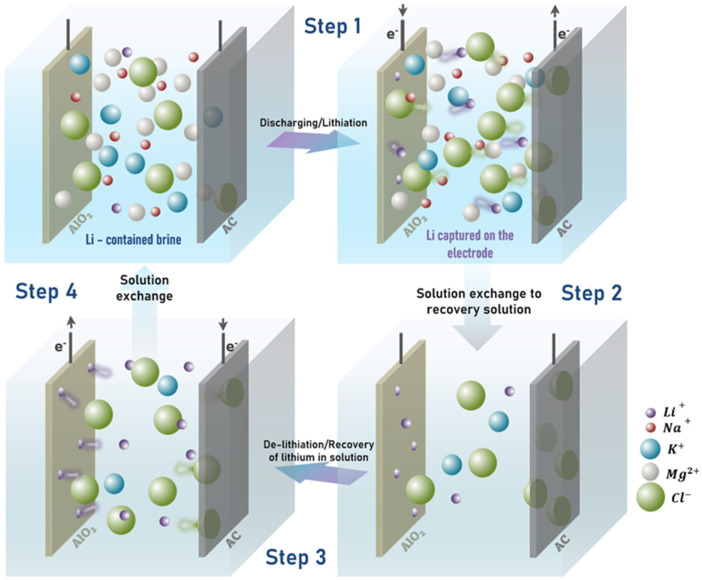
Schematic illustrating the four steps of the lithium recovery cycle from brines using LiAlO_2_ (AlO_2_*) as Li^+^ capturing electrode and activated carbon (AC) counter electrode.

**Figure 3 nanomaterials-13-00895-f003:**
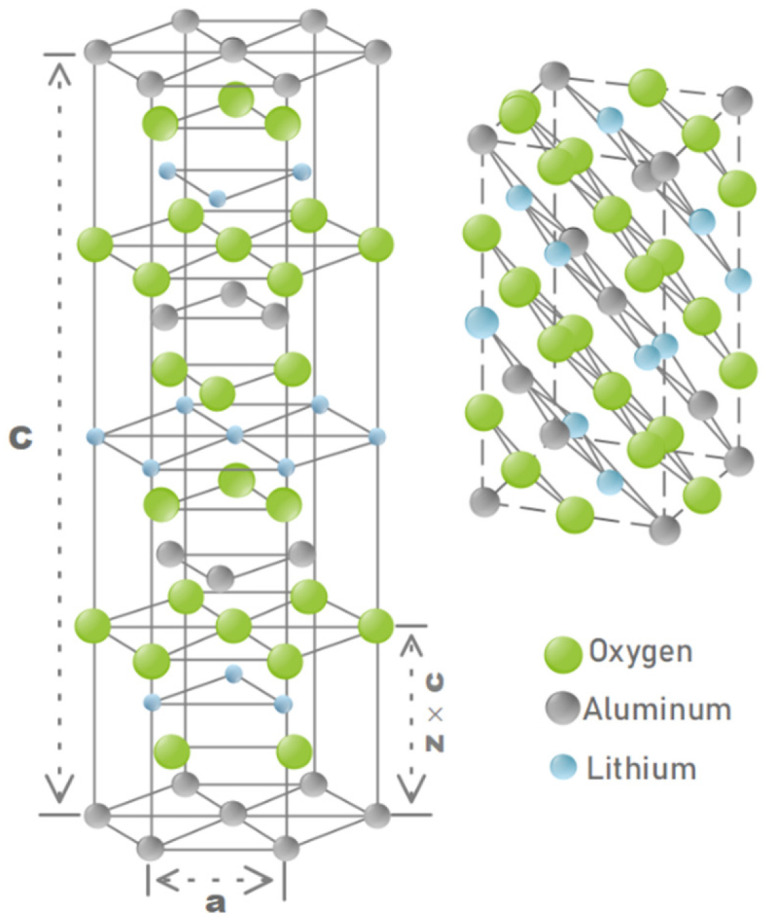
α-NaFeO_2_-type structure of LiAlO_2_.

**Figure 4 nanomaterials-13-00895-f004:**
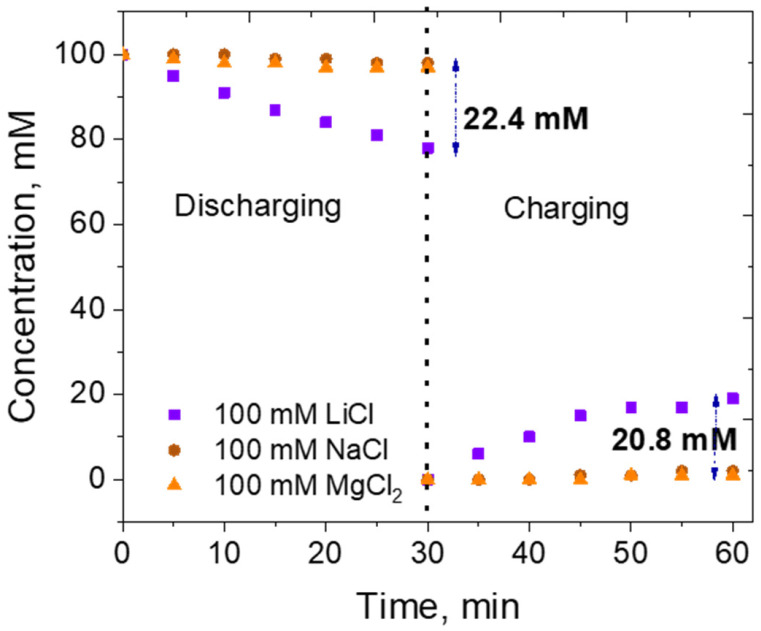
Concentration of the targeted ion in the source solution during the discharging phase, performed at a current density of −0.5 mA cm^−2^ for 30 min, and the concentration of the same ions in 50 mM KCl recovery solution during the charge phase at a current density of 0.5 mA cm^−2^ for 30 min.

**Figure 5 nanomaterials-13-00895-f005:**
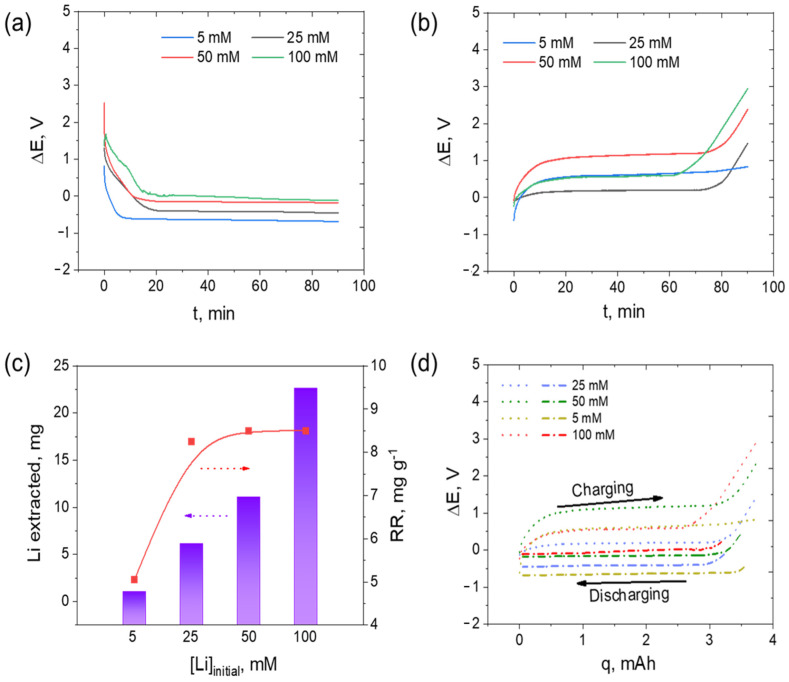
Voltage profile of the cell: (**a**) during Li capture from LiCl solutions of different Li concentrations ranging from 5 mM to 100 mM (Step 1 in Figure 2), (**b**) during the release of Li from (**a**) to a 50 mM KCl recovery solution, (**c**) the amount of Li extracted and the Li recovery rate, and (**d**) the energy cycle in the ΔE vs. q plot for each Li initial concentration condition.

**Figure 6 nanomaterials-13-00895-f006:**
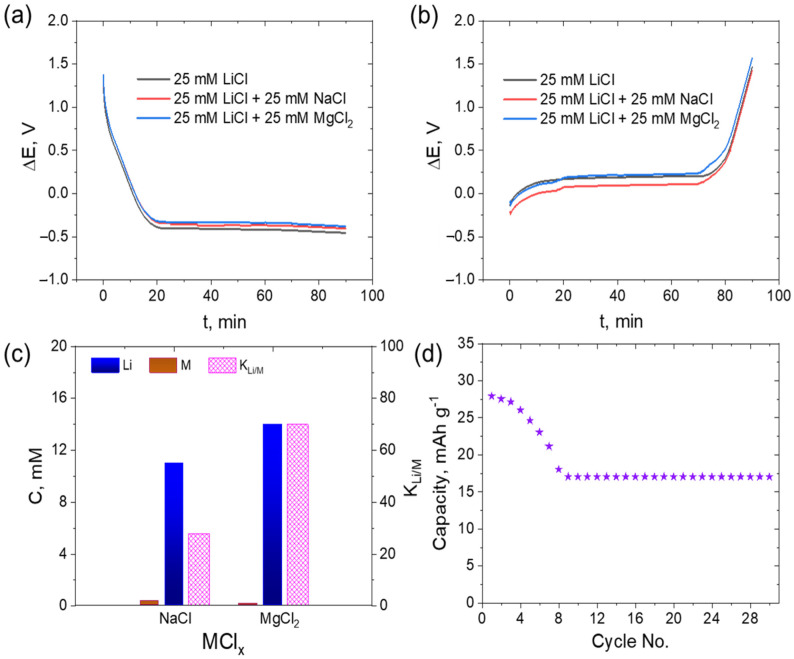
Voltage–time profile of (**a**) discharge cycle and (**b**) charge cycle of LiCl and LiCl containing NaCl or MgCl_2_. (**c**) Li^+^, Na^+^, and Mg^2+^ concentrations after full cycle operation and the respective selectivity (K_Li/M_). (**d**) Capacity over 30 cycles of operation using a 25 mM LiCl salt solution at a current density of ±0.5 mAcm^−2^.

**Table 1 nanomaterials-13-00895-t001:** Chemical composition of a simulated 1st pass SWRO brine operating at 50% efficiency.

Ion	Concentration		(M)/(Li)	
	mg L^−1^	mol L^−1^	wt. Ratio	mol Ratio
Li^+^	0.34	0.049	1	1
Na^+^	24,840	1080	73,058.8	22,040.8
Mg^2+^	540	22.5	1588.2	459.2
K^+^	900	23.1	2647.1	471.4
Ca^2+^	250	6.25	735.3	127.6
Cl^-^	40,720	1147.04	119,764.7	23,409
SO_4_^2-^	580	6.04	1705.9	123.3

## Data Availability

The data presented in this study are available upon request from the corresponding author.

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
