# Peer review of "Novel LiAlO2 Material for Scalable and Facile Lithium Recovery Using Electrochemical Ion Pumping"

_nanomaterials, 2023, doi:10.3390/nano13050895_

Round 1

Reviewer 1 Report

I recommend the article “Novel LiAlO2 material for scalable and facile lithium recovery by electrochemical ion pumping” for publication. The article is interesting and raises current problems of modern development. The authors outlined the research problem well. In addition, the whole is enriched with good graphics to help understand the content. An additional advantage is a possibility of developing research on an industrial scale. The study is in line with current trends, and the resulting electrodes can be durable and environmentally friendly. I think it's a valuable, well-prepared article.

Author Response

Dear Reviewer, we are grateful for your positive feedback and support for publication in this journal

Reviewer 2 Report

The authors developed a hybrid supercapacitor using a LiAlO2-based positive electrode and an activated carbon negative electrode and various aqueous LiCl solution. AlO2/activated carbon electrode allow Li ions to insert and extract. This system is promising for seawater reverse osmosis. This work can be published after minor revision.

1.     The author claims this work can be used for seawater osmosis. However, I do not see relative experiments. Is should be provided.

2.     The capacitance of LiAlO2-based supercapacitor should be provided.

3.     How about the cyclic stability of the supercapacitor?

4.     Whether the LiAlO2 can be used in lithium-ion battery?

5.     The advantages of using aqueous electrolyte should be present in introduction section (Advanced Energy Materials 12 (37), 2202068; Nano-Micro Letters 14 (1), 205; 10.1039/D2EE03749A)

Reviewer 3 Report

Comments:

In this manuscript, the authors reported the development of a hybrid supercapacitor system using α-LiAlO2-based positive electrode and an activated carbon negative electrode to recover Li from aqueous Li resources. The study is comprehensive. I would like to recommend its publication before below few questions are addressed.

1.       Fig. 1b should add a selected area diffraction pattern to show the crystallite of α-LiAlO2. In addition, the caption of Fig. 1b should be revised to make it clear.

2.       The subscript and superscript of words should be carefully checked and revised.

Reviewer 4 Report

This manuscript reports about an hybrid system for lithium recovery, consisting of an LixAlO2 cathode and a carbon anode, during the lithium ion capturing step of operation, when insertion of lithium ions occurs at the cathode and double layer adsorption at the anode. Next step is lithium ions release in the recovery solution by reversing polarity, followed by new operating cycle, after exchange of solution. The principle of operation is similar to an intercalation capacitive deionization process with an insertion type positive electrode, during charging, then with an inverted operation: release of ions during charging of the cell and capture of ions during discharge under short-circuit. The main difference here is that the capture step is done under polarization (first step).

The topic is of great interest; the idea of using LiAlO2 as the active material for lithium recovery by an electrochemical process is clever, however this work has some serious drawbacks.

There is already an extensive literature on electrochemical processes for lithium recovery, see ref. 15 in this manuscript, though the lithium aluminate has not been used before, as far as I can say.

However, the general impression is that the performance of the system presented by the authors are not promising, with the exception of selectivity over magnesium, which is a well-known issue with intercalation-based processes of lithium recovery. Notwithstanding this aspect, the work should be improved in several resepcts.

The characterization and identification of the active material is poor. The authors speak of a layered oxide, but there is no evidence of this in the paper.

Energy consumption is about 4 Wh per gram of lithium; this is a relatively large value compared to other similar and electrochemical processes, see for example in ref. 15.

The cycle life is apparently poor.

In general, there is hardly any discussion and comments of the results.

All these points should be addressed to make this manuscript acceptable, along with specific comments that follow.

Specific comments

Line 22: the expression “…  intercalate Li into the moiety …” is not appropriate. Please revise.

The areal mass of electrode is missing.

Sub-section 3.1: There is no evidence of the formation of a layered oxide.

Figure 1 c: it is not clearly what it is shown in the figure; caption is unclear.

Lines 195-197: the following statement is wrong “The experiment showed the capacitor behavior to form an electrical double layer in 1 M LiCl electrolyte but not in 1 M Na2SO4, indicating a greater Li+ ion conductivity and capacitance by LiAlO2 cathode material compared to Na+ ion”.

First and generally speaking, there is always a double layer at the surface of the electrode, then claiming that such DL forms in one electrolyte and not in another one is a meaningless statement.

Briefly, the interpretation of the cyclic voltammetry curve for the LixAlO2 paste electrode is obviously wrong. Reader would expect the curve to give evidence about insertion and deinsertion of lithium, according to the suggestion and/or assumption of the authors. There appears to be, possibly, an intercalation peak (the peak in the cathodic scan) but not the corresponding de-intercalation peak, in the anodic scan.  

Section 3.2 and throughout manuscript: avoid using formula AlO2, it is meaningless chemically.

Line 332: what is the use of equation 2?

Round 2

Reviewer 4 Report

I am afraid the authors did not address in a convincing way the issues raised in the previous review report.

The following points are in need of authors’ attention and explanations:

(1) According to data reported in the manuscript, namely: XRD pattern in figure 1c, the lithium aluminium oxide has probably a two-phase structure of the γ-LiAlO2 and α-LiAlO2 types. By the way, the diffraction angle range should start below 20° 2θ to catch the main peak of the α-LiAlO2, which is at about 19°.

(2) The claim of the authors to have obtain the α-LiAlO2 is questionable also based on the morphology, as shown by electron microscopy images in figure 1a-b, where there is no evidence of a layered structure. In particular, SEM images show rather thick sheets of material; high resolution transmission electron microscopy is needed to provide morphological / structural evidence of the presence of the layered α-LiAlO2 allotrope.

(3) Authors should explain how the cyclic voltammetry behaviour of the electrode, namely the strong irreversibility of lithium insertion, can be reconciled with the observed capture / release behaviour of the electrode. Actually, looking at the cyclic voltammetry, one can see a much stronger capture than release of lithium ions, which cannot be reconciled with the results of figure 4, among the other.

As a peculiar feature of the CV curve, authors should comment on the presence in the cathodic scan of a plateau of current.

What was the state of the electrode used for cyclic voltammetry? Is this a fresh made electrode or a conditioned electrode? and, if this is the case, what this conditioning consist of?

Did the authors perform cycling voltammetry on other electrodes (in a different state than the electrode in figure 1d) and/or at different (lower) scan rate?

4) equation (2) must be reported where it is referred to; besides, since this is common knowledge, it could be dropped.

(5) to the request of providing areal mass of electrode, the authors replied as follows: “The positive electrode, ready for use in the electrochemical ion pumping cycle, had a weight between 400 mg to 2 g.”

The question is about the areal mass, i.e. the mass of the electrode active material (but the overall mass is fine as well, including all components, LiAlO2, PVDF, and acetylene carbon, since composition is given) per unit area of electrode, in milligram of active material for square centimetre, as it is usually provided.
